# Hyperinflammatory Immune Response in COVID-19: Host Genetic Factors in Pyrin Inflammasome and Immunity to Virus in a Spanish Population from Majorca Island

**DOI:** 10.3390/biomedicines11092548

**Published:** 2023-09-16

**Authors:** Natalia Martínez-Pomar, Vanesa Cunill, Marina Segura-Guerrero, Elisabet Pol-Pol, Danilo Escobar Oblitas, Jaime Pons, Ignacio Ayestarán, Patricia C. Pruneda, Inés Losada, Nuria Toledo-Pons, Mercedes García Gasalla, Joana Maria Ferrer Balaguer

**Affiliations:** 1Immunology Department, Hospital Universitari Son Espases, 07120 Palma de Mallorca, Spain; vanesa.cunill@ssib.es (V.C.); marina.segura@ssib.es (M.S.-G.); juanam.ferrer@ssib.es (J.M.F.B.); 2Health Research Institute of the Balearic Islands (IdISBa), 07120 Palma de Mallorca, Spainialosada@hsll.es (I.L.); nuria.toledo@ssib.es (N.T.-P.);; 3Intensive Care Unit (ICU), Hospital Universitari Son Espases, 07120 Palma de Mallorca, Spain; 4Dreamgenics, 33011 Oviedo, Spain; patricia.cueto@dreamgenics.com; 5Internal Medicine, Hospital Universitari Son Llàtzer, 07198 Palma de Mallorca, Spain; 6Pneumology Department, Hospital Universitari Son Espases, 07120 Palma de Mallorca, Spain; 7Internal Medicine, Hospital Universitari Son Espases, 07120 Palma de Mallorca, Spain

**Keywords:** hyperinflammatory immune response, Pyrin, COVID-19, primary immunodeficiency, cytokine storm, genetic markers

## Abstract

The hyperinflammatory response caused by SARS-CoV-2 infection contributes to its severity, and many critically ill patients show features of cytokine storm (CS) syndrome. We investigated, by next-generation sequencing, 24 causative genes of primary immunodeficiencies whose defect predisposes to CS. We studied two cohorts with extreme phenotypes of SARS-CoV-2 infection: critical/severe hyperinflammatory patients (H-P) and asymptomatic patients (AM-risk-P) with a high risk (older age) to severe COVID-19. To explore inborn errors of the immunity, we investigated the presence of pathogenic or rare variants, and to identify COVID-19 severity-associated markers, we compared the allele frequencies of common genetic polymorphisms between our two cohorts. We found: 1 H-P carries the likely pathogenic variant c.887-2 A>C in the *IRF7* gene and 5 H-P carries variants in the *MEFV* gene, whose role in the pathogenicity of the familial Mediterranean fever (FMF) disease is controversial. The common polymorphism analysis showed three potential risk biomarkers for developing the hyperinflammatory response: the homozygous haplotype rs1231123A/A-rs1231122A/A in *MEFV* gene, the *IFNAR2* p.Phe8Ser variant, and the *CARMIL2* p.Val181Met variant. The combined analysis showed an increased risk of developing severe COVID-19 in patients that had at least one of our genetic risk markers (odds ratio (OR) = 6.2 (95% CI) (2.430–16.20)).

## 1. Introduction

Coronavirus disease 2019 (COVID-19) is characterized by pneumonia, fever, cough, and occasional diarrhea [1]. The disease presents a wide variability of clinical manifestations, from asymptomatic individuals to critical patients with fatal outcomes. Older age, male sex, and the presence of certain comorbidities (e.g., dyslipidemia, obesity, diabetes, cardiovascular disease…) have been identified as predictors of poor outcomes [2]. An inflammatory response to the virus contributes to the disease severity of COVID-19, with approximately 6% of patients progressing to acute respiratory distress syndrome.

Critically ill COVID-19 patients often showed features suggestive of cytokine storm syndrome (CSS) [2,3]. CSS can be associated with several acquired or genetic diseases where the dysfunction can reside at any level of the cellular pathways. Monogenic autoinflammatory diseases, immune defects that predispose to viral infection, and immune dysregulation diseases are three groups of primary immunodeficiencies (PIDs) that predispose to CS. In autoinflammatory syndromes, the production of proinflammatory cytokines occurs in the absence of any trigger due to hyperactivity of innate immunity, and most of the causative genetic mutations lead to dysregulated inflammasome activity. Patients with susceptibility to viral infection and immune dysregulation genetic disorders show impaired viral control, thus resulting in an incomplete viral clearance that causes prolonged stimulation of the immune system and an uncontrolled release of cytokines [4,5,6].

The exact pathogenesis of secondary CSS is not well understood. Several studies found a high rate of heterozygous mutations in genes associated with primary hemofagocytic lymphohistiocytosis (HLH) in patients that developed secondary CSS. These findings have prompted the consideration of the secondary CSS as a T cell-mediated syndrome in which lymphocyte cytolytic dysfunction results in an amplification of the proinflammatory cytokine cascade, leading to severe tissue damage [7,8]. However, different genetic studies showed that other genetically disrupted pathways result in a similar CSS [9]. Chinn et al. identified pediatric patients who met the HLH-2004 criteria with genetic conditions related to inflammasome activity and viral control infection [9,10].

Uncontrolled hyperinflammation caused by SARS-CoV-2 infection leads to the activation of immunity at the multiorgan level and triggers CSS [11]. Host genetic variation influences the development of critical care requiring illness following SARS-CoV-2 infection. Common variants in the human genome affecting the susceptibility to COVID-19 severity have been successfully identified by Genome-Wide Association Studies (GWAS) [12,13,14]. Zhang et al. identified innate immune inborn errors in 2–3% of critical patients and several groups reported isolated immunodeficiency [15,16,17,18]. However, these variants only explain a small fraction of patients or the association is often related to common non-coding variants. Despite the great effort of the scientific community, the host factors’ contribution to disease development and progression needs to be better understood.

In the present study, we focused on causal genes of PIDs associated with CS, whose functional defect affects signaling pathways involved in inflammation, virus defense, and immune regulation. We took an approach to identify genetic biomarkers using two clinical extreme polarized patient groups of Spanish patients from Majorca Island with SARS-CoV-2 infection: a group of patients with critical hyperinflammatory disease and a control cohort, which was established as a risk group for developing severe disease (on the basis of age over 70) that remained asymptomatic after infection. First, we analyzed if monogenetic immune inborn errors were present in critical patients, and secondly, we identified immune-based genetic biomarkers associated with clinical outcomes. Understanding the biological mechanisms that underlay the hyperinflammatory response against SARS-CoV-2 could also guide towards personalized therapeutic options.

## 2. Materials and Methods

### 2.1. Subject Enrollment

We obtained peripheral blood samples from 92 Spanish patients from Majorca Island with a confirmed SARS-CoV-2 viral RNA polymerase-chain-reaction (PCR) test from nasopharyngeal swabs. All enrolled individuals were infected during the first wave and were not vaccinated with SARS-CoV-2. The study was conducted according to the ethical guidelines of the 1975 Declaration of Helsinki and approved by the Balearic Islands Clinical Research Ethics Committee. Written informed consent to have their clinical and genetic information published in medical or scientific journals was obtained from all subjects.

With the objective of finding genetic risk or protective factors related to hyperinflammatory COVID-19, two cohorts with extreme phenotypes of SARS-CoV-2 infection were defined: (1) a hyperinflammatory patients (H-P) cohort and (2) asymptomatic or mild patients (AM-Risk-P) with a high risk (due to older age) of severe COVID-19 as a control cohort (Figure 1a).

The H-P cohort included 52 patients with severe/critical COVID-19 admitted to Son Espases or Son Llatzer Hospitals (Appendix A). The severity of signs and symptoms developed during hospitalization was categorized as severe (grade 2) or critical (grade 3). Severe disease was established when dyspnea was associated with a ≥30/min respiratory rate or <93% blood oxygen saturation or <300 partial pressure of arterial oxygen to the fraction of inspired oxygen ratio and/or >50% lung infiltrates within 24 to 48 hours from admission; critical disease was established for cases with respiratory failure, septic shock, and/or multiple organ dysfunction or failure. In addition, severe or critical patients were classified as hyperinflammatory based on a score for the diagnosis of reactive hemophagocytic syndrome (HScore) [19] over 100. None of the subjects had features of PIDs before the hyperinflammatory COVID-19 diagnosis.

Serum levels of IL-6, IL-10, IL-1, IL-1Ra, IL-18, and sIL-2Rα (sCD25) were analyzed in 22 H-P patients during the acute phase of COVID-19 (Appendix A).

Forty individuals were included in the AM-Risk-P control cohort that was established as a risk group for developing severe disease (on the basis of age over 70) with patients who were asymptomatic or developed mild disease (Appendix A).

All patients were Caucasian with Spanish ancestry, with the exception of three Hispanic patients.

### 2.2. Next Generation Sequencing (NGS)

We designed a custom multi-gene panel (Agilent SureDesign) covering 24 genes related to different PIDs by the International Union of Immunological Societies Expert Committee [20] (Figure 1b): (1) Autoinflammatory disorders with defects affecting the inflammasome: *MEFV*, *MVK*, *NLRC4*, *NLRP12*, *NLRP1*, and *NLRP3*; (2) defects in immunity with predisposition to viral infection and diseases of immune dysregulation: *IRF7*, *IRF3*, *IFNAR2*, *TLR3*, *TICAM1*, *TBK1*, *UNC13D*, *PRF1*, *STXBP2*, *STX11 SLC7A7 CARMIL2/RLTPR*, *CTPS1*, *CD70*, *RASGRP1*, *XIAP*, *SH2D1A*, and *CD27*. The designed probes for this panel include all exons and exon-intron boundaries.

Genomic DNA isolation was performed using the QIAcube automated system (Qiagen, Hilden, Germany). Library preparation was carried out using the SureSelect^QXT^ Target Enrichment for Illumina Multiplexed Sequencing Kit (Agilent Technologies, Santa Clara, CA, USA) according to the manufacturer’s instructions. The obtained DNA fragments were sequenced on an Illumina MiSeq platform. NGS data were processed using the bioinformatics software HD Genome One (DREAMgenics, Oviedo, Spain license number 7157-PS). Briefly, sequencing reads in FASTQ format were filtered by quality scores and aligned to the human genome reference sequence GRCh38. Variants were annotated by using several databases and in silico tools (Ensembl, dbSNP, ClinVar, gnomAD, ESP, Polyphen, SIFT, and Combined Annotation Dependent Depletion score, among others). All studied variants had a total read depth ≥50 with a minimum of 30 reads with the alternative allele count. To confirm a heterozygous variant, variant alleles were required to be present in more than 20% of mapped reads. The Genome Aggregation Database (gnomAD) (https://gnomad.broadinstitute.org/, accessed on 19 July 2022) was referred to extract the frequencies of variants both in the whole population and the European population.

### 2.3. Inborn Errors Test Study

To identify potential inborn errors of immunity, we evaluated, in both cohorts (H-P and AM-Risk-P), the presence of potential disease-causing variants. For this, variants were selected using the following criteria: (1) located in exon or splicing region; (2) excluded synonymous variants; (3) selected variants with allele frequency < 0.1% in internal and public databases (gnomAD); and (4) predicted to be deleterious by, at least, one in-silico tool (SIFT, Polyphen-2, and CADD, among others). Moreover, the inheritance pattern of the disease has been considered. In order to identify variants with potential relevance in FMF, we also considered variants in the *MEFV* gene with an allele frequency > 0.1% and whose role in the pathophysiology of FMF was controversial. Variants were classified according to American College of Medical Genetics and Genomics (ACMG) guidelines [21].

### 2.4. Statistical Analysis to Identify Potential Genetic Markers

To identify potential risk or protective genetic markers, we studied all single nucleotide polymorphisms (SNPs) located in exon or splicing regions, excluding synonymous and intronic variants. SNPs located in X-linked genes were excluded due to sex bias in both groups. Once selected, we compared their allele frequencies in both homozygous and heterozygous conditions between our two patient cohorts. We used a Fisher test to select the variants with a statistically significant different distribution between our patient cohorts (*p*-value  ≤  0.05).

For our selected autosomal biallelic SNP, we calculated the expected genotype frequency in the general European population using the law of Hardy–Weinberg (Hardy–Weinberg Equilibrium Calculator, scienceprimer.com accessed on 26 July 2022) (HW). Public data from gnomAD browser were used to obtain the allele frequencies p and q. The proportion of subjects with genotypes AA, Aa, and aa should follow the p^2^, 2pq, and q^2^. Tests of HW Equilibrium (E) were commonly performed using a simple Chi-square test (χ^2^).

Additionally, the OR and 95% confidence intervals (95% CI) were calculated using logistic regression to evaluate the association between selected polymorphism and COVID-19 severity risk.

## 3. Results

### 3.1. Patients Cohort: Clinical and Laboratory Data

The hyperinflammatory patient cohort (H-P) had a median age of 60 years [age ranging from 19 to 80] and 41 (75%) were men. None of the patients had prior histories suggestive of immunodeficiency. Comorbidity was present in 77% of patients. The most frequent associated comorbidities found were dyslipidemia (54%), hypertension (42%), obesity (Body Mass Index > 30, 29%), and diabetes (29%) (Appendix A).

All patients required supplemental oxygen and developed severe/critical disease. Impending hyperinflammation in our patient cohort was identified by laboratory markers: cytopenias (anemia and lymphopenia), coagulopathy (low platelet and elevated D-dimer levels), tissue damage/hepatitis (elevated lactate dehydrogenase and aspartate aminotransferase serum levels), and macrophage/hepatocyte activation (elevated ferritin levels). Most of the patients (92%) had a significant lower absolute lymphocyte count and a higher neutrophil/lymphocyte ratio (Appendix A). One hundred percent of patients had an HLH score > 100 (36% of patients had an HLH score between 141–188).

All of them presented a hyperinflammatory cytokine pattern and they exhibited high levels of interleukin (IL)-6 (mean value: 416.1 pg/mL), antagonist interleukin 1 receptor (IL-1RA) (mean value: 261.7 pg/mL), and IL-18 (mean value: 77.4 pg/mL), which are particularly higher in CS.

The asymptomatic/mild COVID-19 cohort (AM-risk-P) had an age range of 70–99, with a median age of 84, and 35% of them were male. Comorbidity was present in 69% of the patients, with the following ones being the most frequently associated: hypertension (63%), dyslipidemia (26%), diabetes (26%), and obesity (Body Mass Index > 30, 20%) (Appendix A).

### 3.2. Identifying Monogenic Inborn Errors of Immunity

First, we analyzed genes causing autoinflammatory diseases. In the H-P cohort, we described five patients with controverted clinical significance variants in the *MEFV* gene. The HP-15 presented the variant p.Arg653His with a minor allele frequency (MAF) of 0.000035 and the p.Ala317Val variant, which was not previously described, and its in silico prediction was benign by Polyphen. The p.Arg653His variant has been observed in several individuals affected with familial Mediterranean fever (FMF), but in silico tools (SNPs&GO, Mutation Taster, SIFT, PolyPhen) predicted a benign outcome for this change. The monoallelic p.Lys695Arg variant (MAF 0.005826) was detected in the HP-37 and HP-44 patients. Although this variant was classified as Uncertain Significance (VUS) using ACMG, in the Infevers database the presence of asparagine in this amino acid position (695) is considered probably pathogenic, and several papers report this variant in FMF patients and consider it as a variant with reduced penetrance (Table 1).

We found the p.Pro369Ser-p.Arg408Gln haplotype in two unrelated patients (HP-3 and HP-18). The role of this haplotype in the FMF is unclear, although its frequency in the population is greater than one percent. There are multiple case reports in which patients with atypical or late-onset FMF are found to have this haplotype along with other *MEFV* variants [22,23,24]. The p.Pro369 residue is conserved across mammals and other organisms, and four out of five computational analyses (PolyPhen-2, SIFT, AlignGVGD, BLOSUM, MutationTaster) suggest a deleterious effect in the protein; however, functional studies have not demonstrated any functional alterations [25] (Table 1).

Strikingly, none of the patients of the AM-Risk-P cohort presented pathogenic or controverted clinical significance variants in the *MEFV* gene.

Next, we focused on loci identified as mutated in patients with predisposition to viral infection. In the H-P cohort, we found one heterozygous predicted loss-of-function variant with MAF < 0.0001 at the locus *IRF7*. Specifically, we report the HP-38 patient carrying the *IRF7* canonical splice site variant c.887-2A>C (rs766015923), the prediction of which is loss-of-function with high confidence (Table 1). This patient had never been hospitalized for other life-threatening viral illnesses. As for the AM-risk-P group, we did not find any pathogenic or likely pathogenic variant that predisposes to viral infection or immune dysregulation.

### 3.3. Identifying Genetic Biomarkers

We identified three genetic variants at three loci (Pyrin *MEFV*, interferon receptor alpha, and beta subunit 2 *IFNAR2*, and the regulator of capping protein and myosin 1 linker 2 *CARMIL2*) overrepresented in our hyperinflammatory severe cohort compared to the AM-Risk-P cohort (Table 2). Furthermore, the variants located in *MEFV* and *IFNAR2* genes displayed an HW disequilibrium (HWD) and we considered them as potential risk markers (Table 2).

Interestingly, in the *MEFV* gene, which causes the FMF disease, we observed a new recessive risk genetic marker. Our results showed a significantly enriched homozygous haplotype, dbSNP rs1231123A/A-rs1231122A/A, in the H-P group compared to the AM-Risk-P cohort (36.5% vs. 15%; *p* < 0.05) (Table 2 and Figure 2a).

This *MEFV* haplotype involves two amino acid changes (p.Asp424Glu- and Gly436Arg, respectively) in exon 8 (ENSE00002320009) of the Pyrin isoform O15553-3 (UniProt ID) resulting from the transcript lacking exon 2.

This homozygous haplotype was significantly enriched in our H-P cohort compared to that expected in the European population according to HWE (36.5% vs. 20%; *p* < 0.05). Moreover, it is associated with an increase in COVID-19 severity risk with an odds ratio [OR] of 3.14 (95% CI) (1.123–8.793) (Table 3 and Figure 2a).

In loci related to diseases of predisposition to viral infection and diseases of immune dysregulation, we identified two monoallelic SNPs, the p.Phe8Ser variant (rs2229207) in the *IFNAR2* gene with 8.2% European MAF (eMAF), which was previously related to COVID-19 severity, and the p.Val181Met (rs117556162) (eMAF: 5%) variant in the *CARMIL2* gene. These variants (rs2229207, rs117556162) turned out to be overrepresented in the H-P group compared to the AM-Risk-P group (28.8% vs. 7.5%; *p* < 0.05 and 13% vs. 0%; *p* < 0.05), respectively (Table 2 and Figure 2a).

The *IFNAR2* p.Phe8Ser variant displayed an HWD. In the H-P cohort, its heterozygous frequency was greater than expected in the European population (28.8% vs. 14.7% HWE; *p* < 0.05). Interestingly, the association between this polymorphism and the risk of COVID-19 severity had an odds ratio (OR) of 4.5 (95% CI) (1.204–16.826) (Table 3 and Figure 2a).

The p.Val181Met variant in *CARMIL2* was not detected in any asymptomatic patient. In fact, we found an underrepresentation of the p.Val181Met_*CARMIL2* variant in the AM-Risk-P cohort compared to what was expected (0% vs. 9.5% HWE *p* < 0.01) (Table 2 and Figure 2a).

### 3.4. Distribution of Risk Variants According to Sex

We explored sex distribution in the two groups of patients depending on the presence of every risk variant.

Interestingly, in the H-P group, we found that the majority of rs1231123A/A-rs1231122A/A_*MEFV* carriers were male (18 of 19 (95%)). However, this bias was not observed in the AM-Risk-P group. Furthermore, our analysis of *IFNAR2* p.Phe8Ser carriers did not show significant differences in sex distribution (Figure 2b).

### 3.5. Combined Analysis of Our Candidate Genetic Markers

When we analyzed the presence of our risk genetic markers that displayed an imbalance of HWE with overrepresentation in the H-P cohort (*IFNAR2*, rs117556162, and *MEFV*, rs1231123 A/A–rs1231122 A/A haplotype), we found that the 57% of patients with severe disease had at least one risk marker, while in the asymptomatic group they were represented only in 20% (Figure 2c). The combined analysis of the presence of at least one of our two genetics biomarkers related to inflammasome activity (*MEFV*) and antiviral immunity (*IFNAR2*) was associated with an increase in COVID-19 severity risk with an OR  of 6.274 (95% CI) (2.430–16.201) (Table 3).

## 4. Discussion

Most patients with life-threatening COVID-19 manifest features of HLH/CSS with hyperinflammation signs such as pancytopenia, coagulopathy, and/or liver dysfunction; however, their genetic basis remain undefined. The CS observed in COVID-19 could be due to host genetic factors related to different cellular signaling pathways of the immune system. To better understand the biology of COVID-19 and the mechanisms that connect different loci to the disease outcome, we looked for PIDs causal genes with predisposition to the development of CS (Figure 1b).

We identified a new candidate gene related to severe COVID-19. The analyses of six autoinflammatory disease-related genes showed the *MEFV* gene as a candidate risk biomarker locus. In our analysis, 10% of the H-P had at least one variant of conflicting interpretations of pathogenicity: p.Arg653His, p.Ala317Val, p.Lys695Arg, and the p.Pro369Ser-p.Arg408Gln haplotype, which were not observed in the AM-Risk-P cohort (Table 1). Pathogenic variants in this gene cause the FMF disease that leads to an exaggerated inflammatory response through the uncontrolled production of IL-1. The *MEFV* gene shows the highest rate of mutant alleles in some populations (Jews, Arabs, Turks, and Armenians), but COVID-19 surveillance reports indicate that the disease incidence and death rates for COVID-19 are not greater in the populations where mutant *MEFV* alleles predominate [26]. However, our results suggest that some variants suspicious for FMF pathogenicity may predispose to hyperinflammation in response to novel SARS-CoV-2 and contribute to critical illness. Moreover, our case-control study also points to this gene as a risk factor. We identified a significant enrichment of the homozygous haplotype rs1231123A/A-rs1231122A/A in *MEFV* gene in the H-P-cohort, which displayed a HWD (Table 2).

A SARS-CoV-2 ORF3a protein can provoke a CS by activation of the NLRP3 inflammasome [27]. Pyrin and NLRP3 inflammasomes may compete for an apoptosis-associated speck-like protein containing a caspase recruitment domain (CARD). Pyrin isoform 1 is the main protein, but at least four alternatively spliced transcripts of *MEFV* gen become translated into protein isoforms [28]. *MEFV* haplotype rs1231123A-rs1231122A involves two amino acid changes (p.Asp424Glu and Gly436Arg, respectively) in the exon 8 (ENSE00002320009) of the Pyrin isoform ID: O15553-3 (UniProt) resulting from the transcript lacking exon 2. Several studies revealed that alternatively spliced *MEFV* transcripts lacking exon 2 levels were significantly higher in the leukocytes of FMF patients [29]. In addition, some studies have shown that Pyrin lacking exon 2 shows a different localization pattern, that it is concentrated in the nucleus, and that it may play a relevant role in the inflammation regulation response [29,30,31]. Regarding our H-P cohort, the overrepresentation of the *MEFV* haplotype rs1231123A/A-rs1231122A/A suggests that it is a novel recessive risk factor in the development of the hyperinflammatory COVID-19 disease that appears to be related to male sex (Figure 2b). However, it should be noted that although this haplotype is common in the population, it has not been previously associated with other diseases.

In agreement with other groups [15,32], we found defects in interferon-dependent immunity genes in patients who progressed to hyperinflammation. We identified one patient carrying the *IRF7* canonical splice site variant c.887-2A>C (rs766015923) whose in silico prediction is loss-of-function and clinical significance is probably pathogenic according to ACMG guidelines. The patient was a 52-year-old male that had never been hospitalized for other life-threatening viral illnesses. The Interferon regulatory factor 7 deficiency was described as having an autosomal recessive mode of inheritance [33,34]; however, some studies raise the possibility that the *IRF7* gene exhibits an autosomal dominant inheritance with incomplete penetrance [15].

Different reports have demonstrated the essential role of type I IFN cell-intrinsic immunity in the control of SARS-CoV-2 infection in a similar way as it happens with influenza virus susceptibility [17,33,34]. The rs223657 and rs2834158 variants in the *IFNAR2* gene previously associated with COVID-19 mortality [35] did not show a different distribution between our two patient cohorts. However, in accordance with previous findings, the frequency of the homozygous genotype of the SNP (rs1051393) is higher in our H-P cohort, although the difference was not statistically significant (*p* = 0.087).

Recently, the association of the common variant p.Phe8Ser (rs2229207) affecting interferon receptor gene *IFNAR2* to severe COVID-19 has been proposed in several case-control studies [32,36]. Our data are consistent with previous reports because we found an increased allele frequency of this common variant in H-P patients (Table 2 and Figure 2a) with an HW imbalance. Based on our results, patients who present this SNP had 3.14-fold odds of developing severe COVID-19. This finding and the previously documented association with the outcome and persistence of hepatitis B virus infection [37] suggest that this amino acid change might play a role in type I interferon immunity. The change of the hydrophobic amino acid phenylalanine (F) to the hydrophilic amino acid serine (S) is a functional variant. Studies have shown that IFN-α-induced antiviral protection is significantly increased in cells transfected with the IFNAR2-Phe8 allele compared with the IFNAR2-Ser8 allele [37]. Furthermore, Smieszek et al. [32] described that IFNAR2-Ser8 carriers showed lower levels of IFN-γ. All these pieces of evidence support the consideration of the monoallelic or biallelic IFNAR2-Ser8 variant as a robust risk marker of severe COVID-19.

Male sex is known to be associated with COVID-19 severity. Interestingly, our H-P cohort showed that the 95% of the *MEFV* risk marker carriers were male (Figure 2b). Thus, this genetic marker seems to be a relevant risk factor mainly in males. In contrast, carriers of the genetic marker related to host antiviral defense (*IFNAR2* p.Phe8Ser) did not show a sex bias. Therefore, although these are preliminary results from a limited number of cases, the homozygous *MEFV* haplotype rs1231123A-rs1231122A could be a predictor of disease severity in males.

The *CARMIL2* gene encodes the Capping protein, Arp2/3, and myosin-I linker protein 2, an essential cytosolic protein for the CD28 co-stimulation in T cells that is necessary for IFN-γ production and viral replication inhibition [38]. Defects in this gene cause an autosomal recessive primary immunologic disorder characterized by susceptibility to the Epstein–Barr virus [20,39]. The common rs117556162 (*CARMIL2*, p.Val181Met) was not detected in any individual from the asymptomatic cohort despite the fact that its expected genotype frequency is 9.5% (HW principle). These data make us consider the wild-type residue Valine 181 as a protective factor. The evidence of this residue suggests that it could be a functional variant because it is highly conserved between species, and the missense in silico prediction was detrimental (Polyphen, probably damaging; SIFT, deleterious; Combined Annotation Dependent Depletion score, 25), although neither functional studies nor association with disease has been described in the literature.

Our study has some limitations: the number of subjects enrolled was small, and they were from a specific geographic area (Majorca, Spain’s Mediterranean island). It is known that ancestry genetic markers exhibit substantially different frequencies among populations from different geographic regions. Therefore, the results of the expected genotypic frequencies according to HWE using the allele frequency of the European population may not be conclusive. However, the differences reported between our two cohorts (both from Majorca), despite the small sample size, can provide reliable information. Another important limitation is that the study has only focused on 24 genes related to three signaling pathways: inflammation, virus host defense, and immune regulation. Therefore, neither all the genes related to these pathways nor other PIDs genes are being evaluated in our study.

In conclusion, our results, limited to the Spanish cohort from Majorca, show that the Pyrin inflammasome regulation and antiviral host defense by type I IFN immunity are involved in COVID-19 disease severity. Our data point to the fact that male carriers of the homozygous *MEFV* haplotype rs1231123A-rs1231122A and/or independently gender carriers of the *IFNAR2* p.Phe8Ser variant may tilt the balance toward a hyperinflammatory response after SARS-CoV-2 infection.

## Figures and Tables

**Figure 1 biomedicines-11-02548-f001:**
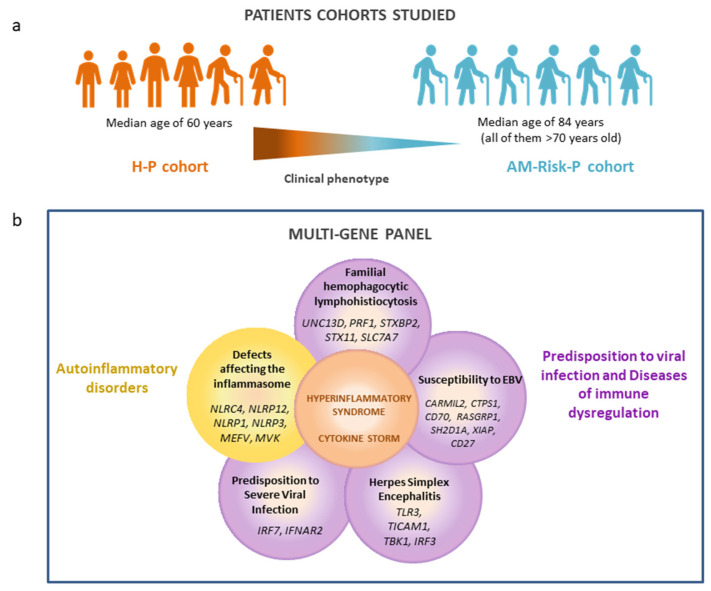
Experimental methodology. (**a**) Cohort of patients: (1) the hyperinflammatory patient cohort H-P included 52 patients that showed hyperinflammatory characteristics according to a score for the diagnosis of reactive hemophagocytic syndrome (HScore) over 100; (2) the asymptomatic or mild patients cohort (AM-Risk-P) included 40 individuals at high risk (based on age greater than 70 years) for severe COVID-19 who remained asymptomatic or developed mild disease that we used as control group. (**b**) Multi-gene panel covering 24 genes related to different PIDs that predispose to cytokine storm distributed in two groups: autoinflammatory disorders or defects in immunity with predisposition to viral infection and diseases of immune dysregulation.

**Figure 2 biomedicines-11-02548-f002:**
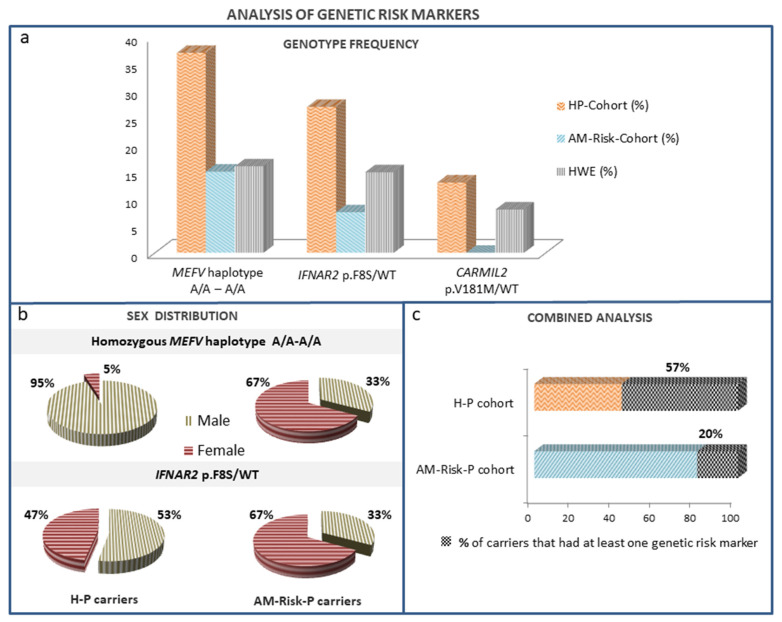
Analysis of genetic risk markers (homozygous *MEFV* haplotype rs1231123A/A-rs1231122 A/A and *IFNAR2* p.F8S/WT). (**a**) Genotype frequency in the patient cohorts (HP-Cohort and AM-Risk-Cohort) compared with the expected genotype frequency according to Hardy–Weinberg Equilibrium (HWE). (**b**) Distribution of carriers according to sex. (**c**) Percentage of carriers that had at least one genetic risk marker in both patient cohorts.

**Table 1 biomedicines-11-02548-t001:** Uncommon/controversial significance variants in H-P cohort. ACMG: American College of Medical Genetics and Genomics, ID: identification number, MAF: minor allele frequency, ND: not previously described, VUS: variant of uncertain significance, WT: wild type.

Patient	Gene	Genotype	ID dbSNP	MAF	ACMG Classification	Pathway
HP-38	*IRF7*	c.887-2A>C/WT	rs766015923	5.95 × 10^−6^	Likely pathogenic	Predisposition to viral infection
HP-15	*MEFV*	p.Arg653His/WT	rs104895085	3.54 × 10^−5^	VUS	Autoinflammatory disorders
	*MEFV*	p.Ala317Val/WT	ND	-	VUS	Autoinflammatory disorders
HP-37	*MEFV*	p.Lys695Arg/WT	rs104895094	5.83 × 10^−3^	VUS	Autoinflammatory disorders
HP-44	*MEFV*	p.Lys695Arg/WT	rs104895094	5.83 × 10^−3^	VUS	Autoinflammatory disorders
HP-3	*MEFV*	p.Arg408Gln/WT	rs11466024	1.34 × 10^−2^	VUS	Autoinflammatory disorders
	*MEFV*	p.Pro369Ser/WT	rs11466023	1.47 × 10^−2^	VUS	Autoinflammatory disorders
HP-18	*MEFV*	p.Arg408Gln/WT	rs11466024	1.34 × 10^−2^	VUS	Autoinflammatory disorders
	*MEFV*	p.Pro369Ser/WT	rs11466023	1.47 × 10^−2^	VUS	Autoinflammatory disorders

**Table 2 biomedicines-11-02548-t002:** Hardy–Weinberg (HW) disequilibrium analysis of the genetic risk factor. The expected genotype frequency in European population (HWE %) was calculated using the eMAF (minor allele frequency in European population) and the HW principle. The *p* (H-P) and *p* (AM-Risk-P) values represent the statistical significance of HW analysis. NS: not statistically significant. WT: wild type.

	Patient Cohort			Hardy–Weinberg Disequilibrium Analysis			
Gene Genotype	H-P (%)	AM-Risk-P (%)	*p*	eMAF	HWE (%)	*p* (H-P)	*p* (AM-Risk-P)
*MEFV* Haplotype rs1231123A/A-rs1231122 A/A	36.5	15	<0.05	0.45	20.3	<0.05	NS
*IFNAR2* p.Phe8Ser/WT	28.8	7.5	<0.05	0.08	14.7	<0.05	NS
*CARMIL2* p.Val181Met/WT	13	0	<0.05	0.05	9.5	NS	<0.01

**Table 3 biomedicines-11-02548-t003:** Association of genetic markers with risk of developing severe COVID-19. The combined analysis included the presence of at least one of our genetic markers (*MEFV* and *IFNAR2*).

Risk Markers	OR	95% CI
*MEFV* Haplotype rs1231123A/A-rs1231122 A/A	3.143	1.123–8.793
*IFNAR2* Phe8Ser/WT	4.500	1.204–16.826
Combined analysis	6.274	2.430–16.201

## Data Availability

The data presented in this study are available upon request from the corresponding author. The data are not publicly available because new data are being analyzed for protective genetic factors.

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
