# Peer review of "Hyperinflammatory Immune Response in COVID-19: Host Genetic Factors in Pyrin Inflammasome and Immunity to Virus in a Spanish Population from Majorca Island"

_biomedicines, 2023, doi:10.3390/biomedicines11092548_

Round 1

Reviewer 1 Report

Authors Martinez-Pomar et al. have submitted a manuscript describing a study which seeks to explain the susceptibility to Cytokine Storm Syndrome (CSS) related to COVID-19 infection in patients.  The authors have taken a unique approach in analyzing genes associated with primary immunodeficiencies as a lens to focus on how COVID-19 responses can vary widely in a population.  The comparisons drawn between severe cases and non-severe is a welcomed addition, especially with HWE included in the analysis. 

The manuscript is well-written, accessible and still timely due to the many unknowns associated with COVID-19 infection, especially in a population that was exposed prior to the availability of vaccination.  

In general, I find little issue with the current study as presented, except for the relatively small and isolated population.  The authors note this in the discussion, but it should be mentioned earlier within the manuscript and qualified as such in the title to note that the study occurred in a very specific population.  

Another option would be for the authors to identify if these variants in the trio of genes are not just statistically supported but in other published databases from COVID patients around the world or in European populations.  I do not think this requires extensive editing or more analyses, per se. 

Essentially, the compelling relationship of the 3 gene variants in ensemble could have a lot of impact in understanding COVID-19 related morbidity and mortality and the authors should broaden the scope of the current study or qualify that the scope is limited to the interesting population in Majorca.  

Author Response

We appreciate reviewers´ critical comments and we believe that they have contributed to improve the quality of our article. Below these lines we address reviewers´ concerns.

  • In general, I find little issue with the current study as presented, except for the relatively small and isolated population.  The authors note this in the discussion, but it should be mentioned earlier within the manuscript and qualified as such in the title to note that the study occurred in a very specific population.  

To address this concern, we have now included the information about the studied population (Spanish patients from Majorca) in the title and the introduction section.

  • Essentially, the compelling relationship of the 3 gene variants in ensemble could have a lot of impact in understanding COVID-19 related morbidity and mortality and the authors should broaden the scope of the current study or qualify that the scope is limited to the interesting population in Majorca.  

Following reviewer 1 suggestion, we have specified in the conclusion paragraph that our results are ”limited to a Spanish cohort from Majorca”.

Reviewer 2 Report

This is a very interesting paper analyzing the role of genetic factors (variants in Pyrin and IFNR genes) in the development of CSS following Sars Cov2 infection. Major limitation to the overall conclusion is the small number of patients as remarked by the authors themselves.

Major issues that need to be improved:

abstract is confusing, mixes results with methods and methods are not well described (the description of two cohorts of patients is missing);

introduction is redundant;

results are too subdivided and the subparagraph headings do not match the contents well.

Author Response

We appreciate reviewers’ critical comments, and we strongly believe that they have also contributed to improve the quality of our article. Following reviewer ’s suggestions, we have made substantial amendments in the manuscript.

Below these lines we address reviewers’ concerns one by one. We hope that after all our modifications and comments, reviewer 2 will consider the article worth of publication

  • Abstract is confusing, mixes results with methods and methods are not well described (the description of two cohorts of patients is missing);

We have expanded and reorganized the abstract section. Methods and results are now clearly separated and described, including the description of our two patients cohort groups: hyperinflammatory patients (H-P) and asymptomatic patients (AM-risk-P).

  • Introduction is redundant;

According to your consideration, introduction section has been summarized and redundant information has been eliminated.

  • Results are too subdivided and the subparagraph headings do not match the contents well.

Following the reviewer's comment, we have eliminated the results subsections by reducing the number of subheadings and changed one title to better summarize the content.